# Role of Cyclosporine in Gingival Hyperplasia: An In Vitro Study on Gingival Fibroblasts

**DOI:** 10.3390/ijms21020595

**Published:** 2020-01-16

**Authors:** Dorina Lauritano, Annalisa Palmieri, Alberta Lucchese, Dario Di Stasio, Giulia Moreo, Francesco Carinci

**Affiliations:** 1Department of Medicine and Surgery, Centre of Neuroscience of Milan, University of Milano-Bicocca, 20126 Milan, Italy; moreo.giulia@gmail.com; 2Department of Experimental, Diagnostic and Specialty Medicine, University of Bologna, via Belmoro 8, 40126 Bologna, Italy; plmnls@unife.it; 3Multidisciplinary Department of Medical and Dental Specialties, University of Campania-Luigi Vanvitelli, 80138 Naples, Italy; alberta.lucchese@unicampania.it (A.L.); dario.distasio@unicampania.it (D.D.S.); 4Department of Morphology, Surgery and Experimental Medicine, University of Ferrara, 44121 Ferrara, Italy; crc@unife.it

**Keywords:** gingival overgrowth, gene expression, drugs, cyclosporine A

## Abstract

Background: Gingival hyperplasia could occur after the administration of cyclosporine A. Up to 90% of the patients submitted to immunosuppressant drugs have been reported to suffer from this side effect. The role of fibroblasts in gingival hyperplasia has been widely discussed by literature, showing contrasting results. In order to demonstrate the effect of cyclosporine A on the extracellular matrix component of fibroblasts, we investigated the gene expression profile of human fibroblasts after cyclosporine A administration. Materials and methods: Primary gingival fibroblasts were stimulated with 1000 ng/mL cyclosporine A solution for 16 h. Gene expression levels of 57 genes belonging to the “Extracellular Matrix and Adhesion Molecules” pathway were analyzed using real-time PCR in treated cells, compared to untreated cells used as control. Results: Expression levels of different genes were significantly de-regulated. The gene *CDH1*, which codes for the cell adhesion protein E-cadherin, showed up-regulation. Almost all the extracellular matrix metalloproteases showed down-regulation (*MMP8*, *MMP11*, *MMP15*, *MMP16*, *MMP24*, *MMP26*). The administration of cyclosporine A was followed by down-regulation of other genes: *COL7A1*, the transmembrane receptors *ITGB2* and *ITGB4*, and the basement membrane constituents *LAMA2* and *LAMB1*. Conclusion: Data collected demonstrate that cyclosporine inhibits the secretion of matrix proteases, contributing to the accumulation of extracellular matrix components in the gingival connective tissue, causing gingival overgrowth. Patients affected by gingival overgrowth caused by cyclosporine A need to be further investigated in order to determine the role of this drug on fibroblasts.

## 1. Introduction

Gingival hyperplasia could occur after the administration of cyclosporine A. It is worsened by bacterial plaque accumulation, and may interfere with normal oral functions such as smiling, speaking, and eating, resulting in psychological problems [1,2]. The incidence of overgrowth of gingiva induced by cyclosporine A (OGIC) is in the range of 20–80% of patients [3]; in addition, up to 90% of transplant recipient patients, who have been submitted to cyclosporine A therapy, present OGIC [4]. Several factors such as age, sex, duration of treatment, and dosage of the prescribed cyclosporine A influence the severity of clinical manifestation of OGIC [5,6,7]. 

Cyclosporine A has an immunosuppressive function, and it is widely administered for autoimmune disease therapy and for treating graft versus host disease in organ transplant patients [8]. To prevent OGIC several approaches have been proposed, such as drug substitution or cyclosporine A dose reduction, as well as oral hygiene programs and surgical interventions, but each of these approaches may have contraindications. To prevent OGIC, reducing the dosage or using alternative drugs is not possible in all situations. Other substitutive drugs have their own side effects too. However, surgical intervention is only proposed for cosmetic cases, whilst oral hygiene protocols have been demonstrated efficacy in controlling OGIC, but could not inhibit its development.

### 1.1. Cyclosporine A and Its Effects on Gingival Fibroblasts Matrix

Literature reported contrasting data about the level of fibrosis recorded in gingival hyperplasia. In the study by Uzel et al., fibrosis has been demonstrated to be low, with a high inflammation degree [9]. However, Kuo et al. stated that all forms of gingival overgrowth are fibrotic [10]. Mechanisms by which OGIC present high inflammation and low fibrosis are unclear. OGIC could result in inflamed gingival tissues with low fibrosis, while kidney fibrosis is a serious consequence of cyclosporine therapy [11,12]. Cyclosporine A may promote inflammation and limit fibrosis due to the innate immune response, and in addiction cyclosporine A may have effects on collagen deposition.

Cyclosporine A may produce an inflammation process with increased fibrotic response in the extracellular matrix. The presence of inflammation may be crucial in order to promote the onset of the fibrotic process, but it is not involved in its development [13]. Several studies stated that pathogen bacteria in gingival tissue as well as mucous membrane trauma may worsen the inflammatory process responsible for the severity OGIC [14]. 

Immune-inflammatory characteristics associated with OGIC show increased macrophage reparative/proliferative phenotypes, variable lymphocyte proportions, and up-regulation of essential growth factors, *IL-1β, and IL-6 cytokines* [15,16]. Some articles have recorded lymphocyte infiltration of plasma cells in OGIC samples of fibroblasts. It can be deduced that the cellular immune response in OGIC fibroblasts is replaced by an immune response, as a consequence of immunosuppressive drugs. Moreover, unlike chronic inflammatory disease, patients with OGIC present a low expression of natural killer lymphocytes. Cell and tissue components of human gingival hyperplasia lesions induced by phenytoin, nifedipine, and cyclosporine A, respectively, have different histological features: phenytoin provokes low inflammation and high fibrosis; nifedipine produces more or less inflammation and fibrosis equally; whilst cyclosporine A produces high inflammation and low fibrosis. OGIC may present a very pronounced activation of the immunity system with some moderate antifibrotic effects in the synthesis and deposition of collagen [9].

A positive equilibrium between the synthesis and degradation of components of the extracellular matrix causes a deposit of the matrix, which has been suggested as one of the most significant events in OGIC [17]. Since the pathogenesis of OGIC is not well known, it is supposed that cyclosporine A influences the regulation of matrix deposition of gingival fibroblasts [18]. 

### 1.2. Objective

In order to demonstrate the effect of cyclosporine A on the extracellular matrix component of fibroblasts, we investigated the expression of 57 genes that code for “Extracellular Matrix and Adhesion Molecules” of gingival fibroblasts.

## 2. Results

The administration of correct concentrations of cyclosporine A is crucial to avoid cellular viability alteration; to determine this, PrestoBlue™ test was performed. Basing on this test, the concentration used for the treatment was 1000 ng/mL.

The gene expression profiles of 57 genes belonging to the “Extracellular Matrix and Adhesion Molecules” pathway were analyzed using real-time PCR (Figure 1). Table 1 reports the list of genes and their fold change. 

Table 2 reports only the significantly deregulated genes. 

Among the most significant up-regulated genes, there was *CDH1* that codes for the cell adhesion protein E-cadherin. 

The E-cadherin levels measured by enzyme linked immunoassay (ELISA) after cyclosporine treatment showed an expected increase of E-cadherin levels (4.5-fold ± 0.3) in treated fibroblasts vs. untreated control (Figure 1), confirming the gene expression results obtained in real-time PCR.

Transmembrane receptors *ITGA2* and *ITGA7* and the basement membrane constituent *LAMB3* presented up-regulation.

Almost all the extracellular matrix metalloproteases (*MMPs*) were characterized by down-regulation (*MMP8*, *MMP11*, *MMP15*, *MMP16*, *MMP24*, *MMP26*). On the contrary *MMP12* and *MMP13* presented up-regulation.

Other genes significantly down-regulated following the treatment with cyclosporine were *COL7A1*, the transmembrane receptors *ITGB2* and *ITGB4*, and the basement membrane constituents *LAMA2* and *LAMB1*. 

Figure 2 shows the significant expression levels of the genes up- and down-regulated in fibroblast cells treated with cyclosporine A.

## 3. Discussion

OGIC is a pathological expression of the consequences of the assault from both mechanical stimuli and side effects of drugs on the oral mucosa. Gingival mucosa has developed biological mechanisms to counteract these trigger factors, and in this process, cells of the innate immune system play a major role. OGIC is the response of molecular regulatory processes that guide the host response to the effect of drug administration and other microbiological patterns. Since innate immunity provides a vigilant mechanism to prevent the fibrotic process of gingival fibroblasts, it is considered to be crucial in the pathogenesis of OGIC. It is thought that up-regulated fibrosis within the cellular microenvironment is one of the key elements causing OGIC; however, how immune and fibrotic processes are regulated and how they may result in different clinical manifestations are initial questions that are beginning to be understood in some detail.

OGIC and its chronic inflammation are sustained by oral biofilm, including bacteria, viruses, and fungi, living in a homeostatic balance with each other and the immune system [17]. Dysbiosis causing OGIC, and the unbalance between oral bacteria and host pro-inflammatory and anti-inflammatory mediators adjunct with fibrotic processes, are factors influencing the severity of OGIC. A weakened immune system as the result from administration of antisuppresant drugs, such as cyclosporine, allows increased dysbiotic oral microflora and sustains the etiopathogenesis of OGIC [18]. Although it is known that microbial dysbiosis promotes OGIC, deregulated immune response mechanisms determine the development and the extent of tissue overgrowth.

The interplay between oral microorganisms and components of the host immune response is responsible for OGIC development. Chemokines and cytokines with their receptors releasing inflammatory mediators play a central role in OGIC development; however, the deregulated tissue homeostasis and inflammation process in OGIC exposes oral mucosa to the products of altered metabolism such as necrotic cells, reactive oxygen, free radicals, and so forth. These host-derived factors could likely alter cells and stimulate an increase of inflammatory mediators, further incrementing inflammation and fibrosis and contributing to OGIC pathogenesis [19]. 

The gingival mucosa is constantly subjected to thermic, chemical, and mechanical insults inducing a permanent state of turnover involving the inflammatory cells, fibroblasts, and inflammatory mediators. Many of these mediators are chemokines and cytokines secreted locally by various cells in the gingiva.

The molecular steps involved in the gingival hyperplasia triggered by cyclosporine A are not well understood [14]. Many factors are involved, such as synthesis and degradation of extracellular matrix molecules, fibroblast proliferation, inhibition of collagen phagocytosis, and decrease of the binding activity in gingival fibroblasts [20]. 

This study used 1000 ng/mL of cyclosporine A in order to stimulate gingival fibroblasts for 16 h. Gene expression analysis showed that only 11 of the analyzed genes were significantly deregulated.

Results obtained showed that cyclosporine A inhibits the secretion of matrix metalloproteases *MMP8*, *MMP11*, *MMP15*, *MMP16*, *MMP24*, and *MMP26* by cultured gingival fibroblasts. The E-cadherin levels measured by enzyme linked immunoassay (ELISA) after cyclosporine treatment showed an expected increase of E-cadherin levels (4.5-fold ± 0.3) in treated fibroblasts vs. untreated control, confirming the gene expression results obtained in real-time PCR. 

*MMPs* contribute to increased synthesis and deposition of collagen and accumulation of extracellular matrix components in the gingival connective tissue, causing gingival overgrowth [18]. 

Other studies about effects of cyclosporine A on gingival fibroblasts showed similar results, indicating a decrease of *MMPs* [21,22]. 

In the gingival overgrowth, fibroblast phagocytosis is responsible for the reduction of collagen and other extracellular matrix molecules. In these mechanisms, many receptors that bind molecules to fibroblasts are involved. Integrins are transmembrane receptors for extracellular matrix molecules, in particular for collagens [18]. 

In our study, two integrins were significantly down regulated: *ITGB2* and *ITGB4*. These findings suggest that cyclosporine A treatment in vitro may reduce collagen phagocytosis inhibiting integrin expression in gingival fibroblasts.

Other integrins like *ITGA2* and *ITGA7* are moderately up-regulated. Literature describes the action of cyclosporine A on the gene expression of extracellular matrix molecules as suppressive or simulative [23]. This variability may result from differences between in vivo vs. in vitro studies. Type, origin, and passage of the cells (in vitro study) and intrinsic differences among subjects (in vivo study) could determine the variability action of cyclosporine A [23,24].

The results of our study demonstrate that cyclosporine may interfere in the modulation of the fibrosis response in gingival fibroblasts, inducing down-regulation of extracellular matrix proteases and, consequently, stimulating deposition of fibrotic tissue. 

## 4. Materials and Methods

### 4.1. Primary Human Fibroblast Cells Culture

Primary gingival fibroblasts, obtained from a woman aged 60 years, were purchased from ATCC® Cell Lines (LGC Standards, Middlesex, UK). Cryopreserved cells at the second passage were cultured in 75 cm^2^ culture flasks containing DMEM medium (Sigma Aldrich, Inc., St Louis, MO, USA) supplemented with 20% fetal calf serum, antibiotics (penicillin 100 U/mL and streptomycin 100 mg/mL-Sigma Aldrich, Inc., St Louis, MO, USA). We have received the informed consent. 

Cell cultures were replicated for subsequent experiments and maintained in water-saturated atmosphere at 37 °C and 5% CO_2_.

### 4.2. Cell Viability Test

A stock solution of cyclosporine A 1 mg/mL was prepared. 

Further dilutions were made with the culture medium to the desired concentrations just before use.

Cells were seeded into 96-well plates at a density of 10^4^ cells per well containing 100 µL of cell culture medium and incubated for 24 h to allow cell adherence. 

Serial dilutions of cyclosporine A (5000, 2000, 1000, 500, and 100 ng/mL) were added (three wells for each concentration). The cell culture medium alone was used as a negative control.

After 24 h of incubation, cell viability was measured using PrestoBlue™ Reagent Protocol (Invitrogen, Carlsbad, CA, USA) according to the manufacturer’s instructions. Briefly, PrestoBlue™ solution (10 µL) was added into each well containing 90 µL of treatment solution. Plates were then placed back into the incubator for 1 h, after which absorbance was measured at wavelengths of 570 nm excitation and 620 nm emission by an automated microplate reader (Sunrise™, Tecan Trading AG, Switzerland). The percentage of viable cells was determined by comparing the average absorbance in drug-treated wells with the average absorbance in control wells exposed to vehicle alone. 

### 4.3. Cell Treatment

Cells were seeded at a density of 1.0 × 10^5^ cells/mL into 9 cm^2^ (3 mL) wells and subjected to serum starvation for 16 h at 37 °C.

Cells were treated with 1000 ng/mL cyclosporine A solution for 24 h. This solution was obtained in DMEM supplemented with 2% FBS, antibiotics, and amino acids. 

Untreated cells cultivated in the presence only of DMEM medium supplemented with 2% FBS, antibiotics, and amino acids were used as control.

The cells were maintained in a humidified atmosphere of 5% CO_2_ at 37 °C. After the end of the exposure time, cells were trypsinized and processed for RNA extraction.

### 4.4. RNA Isolation, Reverse Transcription, and Quantitative Real-Time RT-PCR

Total RNA was isolated from cell lines using GenElute mammalian total RNA purification miniprep kit (Sigma Aldrich, Inc., St Louis, MO, USA) according to manufacturer’s instructions. Pure RNA was quantified with a NanoDrop 2000 spectrophotometer (Thermo Scientific, Waltham, Massachusetts, USA).

cDNA synthesis was performed starting from 500 ng of total RNA using PrimeScript RT Master Mix (Takara Bio Inc., Kusatsu, Japan). The reaction was incubated at 37 °C for 15 min and inactivated by heating at 70 °C for 10 s.

cDNA was amplified by real-time quantitative PCR using the ViiA™ 7 System (Applied Biosystems, Foster City, CA, USA).

All PCR reactions were performed in a 20 µL volume. Each reaction contained 10 µL of 2× qPCRBIO SYGreen Mix Lo-ROX (Pcrbiosystems, London, UK), 400 nM concentration of each primer, and cDNA. 

Custom primers belonging to the “Extracellular Matrix and Adhesion Molecules” pathway were purchased from Sigma Aldrich (Sigma Aldrich, Inc., St Louis, MO, USA). Table 3 reports the primer sequences of genes analyzed. All experiments were performed including non-template controls to exclude reagent contamination. PCR was performed including two analytical replicates. 

The amplification profile was initiated by 10 min incubation at 95 °C, followed by a two-step amplification of 15 s at 95 °C and 60 s at 60 °C for 40 cycles. As the final step, a melt curve dissociation analysis was performed. 

### 4.5. Statistical Analysis

The gene expression levels were normalized to the expression of the reference gene (RPL13) and were expressed as fold changes relative to the expression of the untreated cells. Quantification was done with the delta/delta Ct calculation method [25].

### 4.6. Detection of E-Cadherin Levels by Enzyme-Linked Immunosorbent Assay

E-cadherin levels were measured with sandwich enzyme-linked immunoassay (ELISA) after fibroblast treatment with cyclosporine A by using a commercial kit, Human E-cadherin ELISA KIT (Bioassay Technology Laboratory, Shanghai, China), which uses monoclonal antibodies directed against distinct epitopes of human E-cadherin. 

The plate was pre-coated with human E-cad antibody. Samples were added to these wells and bound to antibodies coated on the wells. Then, biotinylated human E-cad antibody was added and bound to E-cad in the sample. Then, the secondary antibody horseradish peroxidases (HRP)-streptavidin was added and bound to the biotinylated E-cad antibody. After incubation, unbound streptavidin-HRP was washed away during a washing step. Substrate solution was then added, and color developed in proportion to the amount of human E-cad. The reaction was terminated by addition of acidic stop solution, and absorbance was measured at 450 nm by an automated microplate reader (Sunrise™, Tecan Trading AG, Switzerland). E-cadherin levels were expressed as ng E-cad/ng of total protein.

## Figures and Tables

**Figure 1 ijms-21-00595-f001:**
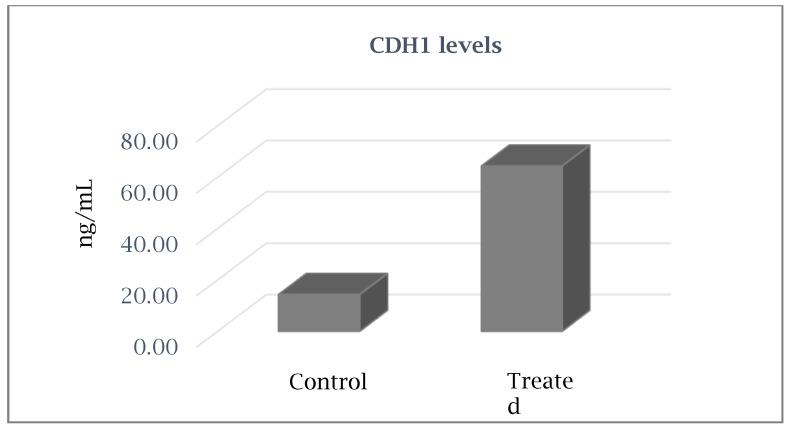
Increased E-cadherin level in fibroblasts after cyclosporine treatment vs. untreated cells (control).

**Figure 2 ijms-21-00595-f002:**
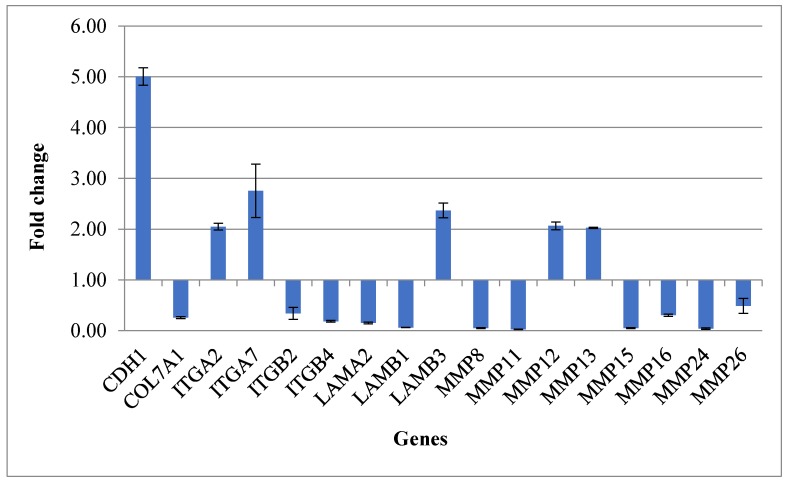
Gene expression profile of fibroblasts treated with cyclosporine A 1000 ng/mL.

**Table 1 ijms-21-00595-t001:** Gene expression profile of 57 genes belonging to the “Extracellular Matrix and Adhesion Molecules” pathway analyzed using real-time PCR. Bold changes represent the expression levels of genes after 24 h treatment with cyclosporine A, as compared with untreated cells.

Gene	Fold Change	Gene Function
*CD44*	0.74	Cell–Cell Adhesion
*CDH1*	**5.01**	Cell–Cell Adhesion
*COL1A2*	0.92	Collagens and Extracellular Matrix Structural constituent
*COL2A1*	0.62	Collagens and Extracellular Matrix Structural constituent
*COL3A1*	0.85	Collagens and Extracellular Matrix Structural constituent
*COL4A1*	1.73	Collagens and Extracellular Matrix Structural constituent
*COL5A1*	0.62	Collagens and Extracellular Matrix Structural constituent
*COL6A1*	0.82	Collagens and Extracellular Matrix Structural constituent
*COL7A1*	**0.26**	Collagens and Extracellular Matrix Structural constituent
*COL8A1*	0.85	Collagens and Extracellular Matrix Structural constituent
*COL9A1*	0.91	Collagens and Extracellular Matrix Structural constituent
*COL10A1*	0.97	Collagens and Extracellular Matrix Structural constituent
*COL11A1*	0.85	Collagens and Extracellular Matrix Structural constituent
*CCTNNA1*	1.25	Cell Adhesion Molecule
*CTNNB1*	1.27	Cell Adhesion Molecule
*CTNND2*	0.96	Cell Adhesion Molecule
*FN1*	0.56	Cell Adhesion Molecule
*HAS1*	0.97	Transmembrane Receptor
*ILF3*	0.93	Transmembrane Receptor
*ITGA1*	1.31	Transmembrane Receptor
*ITGA2*	**2.05**	Transmembrane Receptor
*ITGA3*	1.47	Transmembrane Receptor
*ITGA4*	0.99	Transmembrane Receptor
*ITGA5*	1.51	Transmembrane Receptor
*ITGA6*	1.08	Transmembrane Receptor
*ITGA7*	**2.75**	Transmembrane Receptor
*ITGA8*	0.77	Transmembrane Receptor
*ITGB1*	1.03	Transmembrane Receptor
*ITGB2*	**0.34**	Transmembrane Receptor
*ITGB4*	**0.19**	Transmembrane Receptor
*ITGB5*	0.63	Transmembrane Receptor
*LAMA1*	0.71	Basement Membrane Constituent
*LAMA2*	**0.15**	Basement Membrane Constituent
*LAMA3*	0.89	Basement Membrane Constituent
*LAMB1*	**0.06**	Basement Membrane Constituent
*LAMB2*	1.08	Basement Membrane Constituent
*LAMB3*	**2.37**	Basement Membrane Constituent
*MMP2*	0.77	Extracellular Matrix Protease
*MMP3*	1.05	Extracellular Matrix Protease
*MMP7*	0.78	Extracellular Matrix Protease
*MMP8*	**0.05**	Extracellular Matrix Protease
*MMP9*	0.92	Extracellular Matrix Protease
*MMP10*	1.70	Extracellular Matrix Protease
*MMP11*	**0.03**	Extracellular Matrix Protease
*MMP12*	**2.07**	Extracellular Matrix Protease
*MMP13*	**2.03**	Extracellular Matrix Protease
*MMP14*	0.98	Extracellular Matrix Protease
*MMP15*	**0.05**	Extracellular Matrix Protease
*MMP16*	**0.31**	Extracellular Matrix Protease
*MMP24*	**0.04**	Extracellular Matrix Protease
*MMP26*	**0.49**	Extracellular Matrix Protease
*TGFB1*	1.24	TGFβ Signaling
*TGFB2*	0.74	TGFβ Signaling
*TGFB3*	0.77	TGFβ Signaling
*TIMP1*	0.88	Extracellular Matrix Protease Inhibitor
*VCAN*	0.88	Cell Adhesion Molecule
*RPL13*	1.00	Housekeeping gene

Bold fonts indicate significant variation of gene expression level: fold change ≥ 2 and *p* value ≤ 0.05 for up-regulated genes, and fold change ≤ 0.5 and *p* value ≤ 0.05 for significantly down-regulated genes.

**Table 2 ijms-21-00595-t002:** Significant gene expression levels after 24 h treatment with cyclosporine A, as compared with untreated cells.

Gene	Fold Change	SD (+/−)	Gene Function
*CDH1*	5.01	0.17	Cell–Cell Adhesion
*COL7A1*	0.26	0.02	Collagens and Extracellular Matrix Structural constituent
*ITGA2*	2.05	0.07	Transmembrane Receptor
*ITGA7*	2.75	0.52	Transmembrane Receptor
*ITGB2*	0.34	0.12	Transmembrane Receptor
*ITGB4*	0.19	0.02	Transmembrane Receptor
*LAMA2*	0.15	0.02	Basement Membrane Constituent
*LAMB1*	0.06	0.00	Basement Membrane Constituent
*LAMB3*	2.37	0.14	Basement Membrane Constituent
*MMP8*	0.05	0.01	Extracellular Matrix Protease
*MMP11*	0.03	0.00	Extracellular Matrix Protease
*MMP12*	2.07	0.08	Extracellular Matrix Protease
*MMP13*	2.03	0.01	Extracellular Matrix Protease
*MMP15*	0.05	0.01	Extracellular Matrix Protease
*MMP16*	0.31	0.02	Extracellular Matrix Protease
*MMP24*	0.04	0.01	Extracellular Matrix Protease
*MMP26*	0.49	0.15	Extracellular Matrix Protease

**Table 3 ijms-21-00595-t003:** Primer sequences of genes belonging to the “Extracellular Matrix and Adhesion Molecules” pathway analyzed using real-time PCR.

Gene Name	Primer Sequence 5′–3′
*CD44*	f-TTATCAGGAGACCAAGACC r-ATCCATTCTGGAATTTG
*CDH1*	f-CTGGGCAGAGTGAATTTTG r-GACTGTAATCACACCATCTG
*COL1A2*	f-GTGGTTACTACTGGATTGAC r-CTGCCAGCATTGATAGTTTC
*COL2A1*	f-GAAGAGTGGAGACTACTGG r-CAGATGTGTTTCTTCTCCTTG
*COL3A1*	f-ATTCACCTACACAGTTCTGG r-TGCGTGTTCGATATTCAAAG
*COL4A1*	f-AAAGGGAGATCAAGGGATAG r-TCACCTTTTTCTCCAGGTAG
*COL5A1*	f-TTGACGAGAACTACTACGAC r-ATCCCTTCATAGATGGTATCC
*COL6A1*	f-AGCTCAATGTCATTTCTTGC r-AGGTGTAATCTGGACACTTC
*COL7A1*	f-ATGACCTTGGCATTATCTTG r-TGAATATGTCACCTCTCAAGG
*COL8A1*	f-CTCAAGAAGCTGTTGTGAAG r-CAGGACTGCTGAATCAAATC
*COL9A1*	f-ACCTAAAGGTGACTTGGG r-CATTTCTGCCATAGCTGG
*COL10A1*	f-GCTAGTATCCTTGAACTTGG r-CCTTTACTCTTTATGGTGTAGG
*COL11A1*	f-AGATGAGGCAAACATCGTTGA r-ATCAGAATCCCTGCCGTCTA
*CCTNNA1*	f-CAAACAAATCATTGTGGACC r-CACTAATGATGCTTTCCAGAC
*CTNNB1*	f-CAACTAAACAGGAAGGGATG r-CACAGGTGACCACATTTATATC
*CTNND2*	f-AGAGAATTTGGATGGAGAGAC r-TTGTTGTCTCCAAAACAGAG
*FN1*	f-CCATAGCTGAGAAGTGTTTTG r-CAAGTACAATCTACCATCATCC
*HAS1*	f-TACTTCCACTGTGTATCCTG r-GTGTACTTGGTAGCATAACC
*ILF3*	f-CCTGTGTGAGAAATCCATTG r-TTCACAAGGGTCATAAATGC
*ITGA1*	f-CAGGTTGGAATTGTACAGTATG r-TGTCTATTCCAAGAGCTTGTC
*ITGA2*	f-GGTGGGGTTAATTCAGTATG r-ATATTGGGATGTCTGGGATG
*ITGA3*	f-AGGTAATCCATGGAGAGAAG r-GTAGAAGTTCTCATCCACATC
*ITGA4*	f-AAAGCTTGGATCGTACTTTG r-CTCTTCCTTCCTCTCTGATG
*ITGA5*	f-AAGCTTGGATTCTTCAAACG r-TCCTTTTCAGTAGAATGAGGG
*ITGA6*	f-AAATACCAAACCAACACAGG r-TACTGAATCTGAGAGGGAAC
*ITGA7*	f-CATGAACAATTTGGGTTCTG r-GCCCTTCCAATTATAGGTTC
*ITGA8*	f-CAATATTGGACCAAGTACCATC r-GGCTTTATATCCTGTGGATTG
*ITGB1*	f-ATTCCCTTTCCTCAGAAGTC r-TTTTCTTCCATTTTCCCCTG
*ITGB2*	f-AGATTTGCAGGTATTGATGG r-ATTTCTAAAGCAATAGCCCG
*ITGB4*	f-ATCTGGACAACCTCAAGAAG r-GCCAAATCCAATAGTGTAGTC
*ITGB5*	f-TAGTATCCGGTCTAAAGTGG r-CTCTGACCAGGATAGGATAC
*LAMA1*	f-CATACATCACTCATCAATGGC r-TAGCGTCTGGTAACAATAGG
*LAMA2*	f-AATAAATCTCGCTGTGAGTG r-GTTAGAAAAGTTCCAGCTCTC
*LAMA3*	f-GAATCAGTTGCTCAACTACC r-TCAGTCAGTTCTCTTTCCAG
*LAMB1*	f-GTGTGTATAGATACTTCGCC r-AAAGCACGAAATATCACCTC
*LAMB2*	f-AGTTTCATTTCACACACCTC r-ACTCACAGACTACATCATCC
*LAMB3*	f-CTTATGGATTTAGTGTCTGGG r-CTGGTGAAATTGACTCTCAAG
*MMP2*	f-GTGATCTTGACCAGAATACC r-GCCAATGATCCTGTATGTG
*MMP3*	f-TTTCCCAAGCAAATAGCTGAA r-AGTTCCCTTGAGTGTGACTCG
*MMP7*	f-GGGATTAACTTCCTGTATGC r-GATCTCCATTTCCATAGGTTG
*MMP8*	f-AAGTTGATGCAGTTTTCCAG r-CTGAACTTCCTTCAACATTC
*MMP9*	f-AAGGATGGGAAGTACTGG r-GCCCAGAGAAGAAGAAAAG
*MMP10*	f-AGCGGACAAATACTGGAG r-GTGATGATCCACTGAAGAAG
*MMP11*	f-GATAGACACCAATGAGATTGC r-TTTGAAGAAAAAGAGCTCGC
*MMP12*	f-AGGTATGATGAAAGGAGACAG r-ATCAATTTTAGGCCCGATTC
*MMP13*	f-AGTTCGGCCACTCCTTAGGT r-TGGTAATGGCATCAAGGGAT
*MMP14*	f-ATGGCAAATTCGTCTTCTTC r-CGTTGAAACGGTAGTACTTG
*MMP15*	f-ATCTGACCTTTAGCATCCAG r-CAAAGAGTACCATGATGTCG
*MMP16*	f-ACCCTCATGACTTGATAACC r-TCTGTCTCCCTTGAAGAAATAG
*MMP24*	f-CAAAGGTGACAAGTATTGGG r-TTGAAAAAGTAGGTCTTGCC
*MMP26*	f-AAGGATCCAGCATTTGTATG r-CTTTGATCCTCCAATAAACTCC
*TGFB1*	f-AACCCACAACGAAATCTATG r-CTTTTAACTTGAGCCTCAGC
*TGFB2*	f-AGATTTGCAGGTATTGATGG r-ATTTCTAAAGCAATAGGCCG
*TGFB3*	f-TGTTGAGAAGAGAGTCCAAC r-ATCACCTCGTGAATGTTTTC
*TIMP1*	f-CACCTTATACCAGCGTTATG r-TTTCCAGCAATGAGAAACTC
*VCAN*	f-CCAGTGTGAACTTGATTTTG r-CAACATAACTTGGAAGGCAG
*RPL13*	f-ATTCACAAGAAGGGAGACAG r-GAAATTCTTCTCTTCCTCAGTG

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
