# Peer review of "Role of Cyclosporine in Gingival Hyperplasia: An In Vitro Study on Gingival Fibroblasts"

_ijms, 2020, doi:10.3390/ijms21020595_

Round 1

Reviewer 1 Report

Authors examined the role of cyclosporine A on gingival fibroblast to verify its role in gingival hyperplasia.

The study is well thought and designed and I enjoyed reading it. 

Author Response

Milan 09/01/2020

Dear Editor,

attached please find the revised version of our original manuscript entitled " Role of Cyclosporine in Gingival Hyperplasia: An in Vitro Study on Gingival Fibroblasts". I thank you for enjoying reading our paper.

Best regards

Dorina Lauritano

Reviewer 2 Report

In the manuscript titled “Molecular aspects of drug-induced gingival overgrowth: An in vitro study on cyclosporine and gingival fibroblasts”, in order to determine if cyclosporine altered the responses of ECM component of fibroblasts, the authors investigated the gene expression profile of human fibroblasts after cyclosporine treatment. The results revealed that a few genes were upregulated whereas many were downregulated after cyclosporine treatment.

Following are my suggestions:

There are many grammatical concerns and oddly worded sentences throughout the manuscript. The authors should perform a thorough proof-reading. Page 2, line 75-79…………….”The cellular and tissue features……………………inflammation and low fibrosis”. Please provide a reference for this sentence. Table 1 shows a fold change of 57 genes in comparison to what? The authors performed ELISA to measure the change in gene expression of E-cadherin after cyclosporine treatment. They should show the results in the form of a bar plot. Page 7, line 229……………the primer sequences should be included in the manuscript as a supplementary file. Page 8, line 253-273……………..author contributions, funding source, acknowledgements and conflicts of interest should be added.

Author Response

Milan, 09/01/2020

Dear Editor,

attached please find the revised version of our original manuscript entitled " Role of Cyclosporine in Gingival Hyperplasia: An in Vitro Study on Gingival Fibroblasts". All concerns raised from reviewers were addressed in the text (highlighted in yellow) and in this letter. The language of the manuscript has also been extensively revised by a professional MDPI English language science editing service and all authors of this article have seen and approved the changes.

Best regards

Dorina Lauritano

REVIEWER 2

Reference for the sentence “The cellular and tissue features……………………inflammation and low fibrosis” has been added.

Table 1 legend has been corrected.

ELISA test results have been presented in the form of a bar plot.

Primer sequences have been included in Table III as a supplementary file.

Author contributions, funding sources, acknowledgments and conflicts of interest have been added. 

Two gene names in the table I have been corrected.

Round 2

Reviewer 2 Report

The manuscript has improved after the revision.

This manuscript is a resubmission of an earlier submission. The following is a list of the peer review reports and author responses from that submission.